# Frequency and Predictors of Dysplasia in Pseudopolyp-like Colorectal Lesions in Patients with Long-Standing Inflammatory Bowel Disease

**DOI:** 10.3390/cancers15133361

**Published:** 2023-06-27

**Authors:** Elena De Cristofaro, Elisabetta Lolli, Stefano Migliozzi, Stella Sincovih, Irene Marafini, Francesca Zorzi, Edoardo Troncone, Benedetto Neri, Livia Biancone, Giovanna Del Vecchio Blanco, Emma Calabrese, Giovanni Monteleone

**Affiliations:** 1Azienda Ospedaliera Policlinico Tor Vergata, 00133 Rome, Italy; elena.decristofaro@ptvonline.it (E.D.C.); elisabetta.lolli@ptvonline.it (E.L.); irene.marafini@ptvonline.it (I.M.); biancone@med.uniroma2.it (L.B.); giovanna.delvecchioblanco@ptvonline.it (G.D.V.B.);; 2Department of Systems Medicine, University of Rome “Tor Vergata”, 00133 Rome, Italy; stefano.migliozzi@ptvonline.it (S.M.); stella.m.s@hotmail.it (S.S.);

**Keywords:** Crohn’s disease, ulcerative colitis, colitis-associated colorectal cancer, surveillance colonoscopy

## Abstract

**Simple Summary:**

Inflammatory bowel disease (IBD) patients with long-standing and extensive colitis have an enhanced risk of developing colorectal cancer and, therefore, they must undergo surveillance colonoscopies at regular intervals. The presence of polyps in an area inside colitis, commonly referred to as “pseudopolyps”, seems to increase the risk of CRC in IBD, but it is unclear whether there are some subsets of pseudopolyp-like lesions that might eventually undergo neoplastic transformation. In the present study, we evaluated the frequency and predictors of dysplasia in IBD-associated pseudopolyp-like colorectal lesions. Our study shows that more than one-fifth of pseudopolyp-like lesions are dysplastic, and the majority of such neoplastic lesions are greater than 5 mm and located in the right colon. Therefore, lesions with such morphologic features must be removed even if they are present in an area inside colitis.

**Abstract:**

Current endoscopic surveillance programs do not consider inflammatory bowel disease (IBD)-associated post-inflammatory polyps (pseudopolyps) per se clinically relevant, even though their presence seems to increase the risk of colorectal cancer (CRC). However, it remains unclear whether the link between pseudopolyps and CRC is indirect or whether some subsets of pseudopolyp-like lesions might eventually undergo neoplastic transformation. This study aimed to assess the frequency and predictors of dysplasia in pseudopolyp-like lesions in a population with long-standing colonic IBD. This was a retrospective, single-center study including patients with a colonic IBD (median disease duration of 192 months) and at least a pseudopolyp-like lesion biopsied or resected in the period from April 2021 to November 2022. One hundred and five pseudopolyps were identified in 105 patients (80 with ulcerative colitis and 25 with Crohn’s disease). Twenty-three out of 105 pseudopolyp samples (22%) had dysplastic foci, and half of the dysplastic lesions were hyperplastic. Multivariate analysis showed that the age of the patients (odds ratio (OR) 1.1; *p* = 0.0012), size (OR 1.39; *p* = 0.0005), and right colonic location (OR 5.32; *p* = 0.04) were independent predictors of dysplasia, while previous exposure to immunosuppressors/biologics and left colonic location of the lesions were inversely correlated to dysplasia (OR 0.11; *p* = 0.005, and OR 0.09; *p* = 0.0008, respectively). No differences were seen between ulcerative colitis and Crohn’s disease patients. Lesions with a size greater than 5 mm had a sensitivity of 87% and a specificity of 63% to be dysplastic. These data show that one-fourth of pseudopolyp-like lesions evident during surveillance colonoscopy in patients with longstanding IBD bear dysplastic foci and suggest treating such lesions properly.

## 1. Introduction

Patients with long-standing Crohn’s disease (CD) colitis and patients with long-standing ulcerative colitis (UC) are at increased risk of developing colorectal cancer (CRC) [1,2,3] and, therefore, they must undergo surveillance colonoscopies at regular intervals [4,5,6,7]. The known risk factors associated with dysplasia include age at diagnosis (<30 years) [8], long duration and extent of colitis, presence of colonic strictures, severe chronic colonic inflammation, family history of CRC, and primary sclerosing cholangitis (PSC) [9,10,11]. It has also been reported that patients with polyps in an area of colitis, commonly referred to as “post-inflammatory polyps” or “pseudopolyps”, have an increased risk of colonic dysplasia and CRC. Specifically, the presence of pseudopolyps increases the risk of CRC by 1.9 to 2.5 fold, and the current guidelines suggest colonoscopy at 2- or 3-year intervals considering these patients are at intermediate risk for CRC [9,10,11,12,13]. A possible explanation of the link between pseudopolyps and the risk of CRC lies in the fact that they represent an indirect marker of previous episodes of severe colonic inflammation and their incidence rises with the extent of colitis. It is also plausible that the presence of pseudopolyps, especially when they are very numerous, may make it more difficult for the operator to detect dysplastic lesions while performing surveillance colonoscopy [10,14,15].

Generally, pseudopolyps are multiple, their surface is smooth, and the borders are well-definite, and current endoscopic surveillance programs do not consider pseudopolyps per se clinically relevant because the direct malignant transformation of pseudopolyps is considered rare or unlikely. Nonetheless, several reports documented carcinoma or dysplasia features in pseudopolypoid lesions [16,17,18]. Additionally, even in the same patient, there can be a broad spectrum of endoscopic appearance of the pseudopolyps, and their Kudo pit pattern [19] cannot be always determined, especially by non-expert endoscopists. In case of doubts, pseudopolyp-like lesions need to be removed or biopsied, but little is known about the possibility that such lesions can indeed bear neoplastic foci. Moreover, the clinical and endoscopic factors that could help to identify pseudopolyp-like lesions at high risk of dysplasia remain unknown. 

This study aimed to identify the frequency and the predictors of dysplasia in inflammatory bowel disease (IBD)-associated pseudopolyp-like colorectal lesions in a population with long-standing colonic disease.

## 2. Materials and Methods

### 2.1. Study Population and Data Collection

This monocentric, retrospective study included patients with colonic IBD with evidence of pseudopolyp-like lesions, which were biopsied or resected during a surveillance colonoscopy performed at the “Tor Vergata University” hospital (Rome, Italy) from April 2021 until November 2022. Inclusion criteria were: (1) diagnosis of UC, CD, or IBD-unclassified (IBD-U); (2) confirmed IBD of at least 8 years or any duration in the patients with a concomitant PSC; (3) left-sided or extensive colitis, with involvement of >30% of the colonic surface for CD or IBD-U), or any extent if there was a concomitant PSC; and (4) presence of at least one lesion defined as “post-inflammatory polyps” or “pseudopolyps” by an endoscopist and biopsied or resected when it had an ulcerated surface, was solitary, or had no clear Kudo pit pattern. In case of multiple lesions, the pseudopolyp with the greater size was biopsied or removed. The patients were recruited from the electronic endoscopic dataset using the following terms: CD; UC; IBD-U; and post-inflammatory polyps or pseudopolyps. Demographic and clinical characteristics of the patients were collected from medical records and included sex, age, smoking habits, behavior, and extent of disease using Montreal classification [20], history of medical and current therapies, familiar history of CRC, previous dysplasia, and concomitant PSC. Clinical activity was assessed with a Partial Mayo score for UC and Harvey–Bradshaw Index (BHI) for CD [21]. Surveillance procedures were defined as colonoscopies in which either segmental random biopsies or chromoendoscopy were employed. For each endoscopic procedure, the following data were collected: date of procedure, caecal intubation, disease activity (defined using the Mayo UC score and SES-CD) [5], the extent of the lesions, quality of bowel preparation (defined as excellent or good) or inadequate (defined as fair or poor) using the Boston Bowel Preparation Score (BBPS) [22], and the extent of intubation and technique of mucosa evaluation (i.e., high definition white light, narrow binding imaging (NBI) or dye-chromoendoscopy (DCE) with methylene blue or indigo carmine). The pseudopolyps were defined according to location, size, ulcerated surface, morphology, and number. A description of the surrounding mucosa was also provided. Biopsied or resected pseudopolyps were histologically evaluated and classified as adenomatous, colitis-associated dysplasia, sessile serrated adenomatous/lesions (SSA/L), or hyperplastic. The presence and the grade of dysplasia (low and high grade) were specified, and the indefinite dysplasia was considered negative. The study was approved by the local Ethics Committee (N. 2022/106.22).

### 2.2. Statistical Analysis

Qualitative data were expressed as numbers and proportion (%) and quantitative data were expressed as average and ± standard deviation or median (range). The characteristics of the patients were compared by using χ square test or the exact Fisher test for categoric variables and Mann–Whitney test for continuous variables. Logistic regression was used to identify predictors associated with dysplasia (odds ratio; OR). The parameters with *p* < 0.05 in the univariate analysis were used to perform a multivariate logistic regression analysis to determine their influence on the risk of dysplasia. The receiver operating characteristic (ROC) curve was plotted to identify the lesion’s size and age cut-off. A *p*-value < 0.05 was considered statistically significant. Statistical analysis was performed by using GraphPad Prism version 9.0.

## 3. Results

### 3.1. Patients’ Characteristics

One hundred and five (52%) patients were included in our analysis as they had at least one pseudopolyp-like lesion, which exhibited some endoscopic features that made it difficult to be interpreted. Specifically, 33 lesions had an ulcerated surface, 21 were solitary and 84 were multiple, and 17 had an undeterminable Kudo’s pit pattern. All the lesions were located in an area of colitis, and they were either biopsied (*n* = 96) or removed (*n* = 9). Seventy patients (67%) were male. Eighty patients (76%) had a UC, of which 28 (35%) had left-sided colitis and 52 (65%) a pancolitis. Twenty-five (24%) patients had a CD, of which 6 (24%) had an isolated colonic disease (L2) and 19 (76%) an ileal-colonic disease (L3). No patient had IBD-U. No patient underwent previous colonic surgery. The median disease duration was 192 months (range 90–504), and the median age at enrolment was 53.5 years (range 24–80). Seventeen patients (16%) had a first-degree familiarity with CRC, while in 6 (5%) patients, a dysplastic lesion was diagnosed in previous colonoscopies. Among UC patients, 64 (80%) were in clinical remission or had mild clinical activity (Mayo partial score < 5) at the time of endoscopy, and 16 (20%) patients had moderate/severe clinical activity (Mayo partial score ≥ 5). Among CD patients, 20 (80%) were either in clinical remission or mild activity (Harvey–Bradshaw Index < 8), and 5 (20%) had moderate/severe activity (Harvey–Bradshaw Index ≥ 8). The demographic and clinical characteristics are shown in Table 1.

### 3.2. Endoscopic Characteristics

Of the 105 procedures, 51 (49%) were performed with white light and random biopsies, 16 (15%) with virtual chromoendoscopy (NBI), and 38 (36%) with DCE (36%). Excellent/adequate bowel preparation (BBPS ≥ 6) was documented in 79 colonoscopies (75%). Among the 80 UC patients, the endoscopic activity was absent/mild (0–1 Mayo endoscopic sub-score) in 41 (51%), and moderate/severe (2–3 Mayo endoscopic sub-score) in 39 (49%). Among the 25 CD patients, 15 (60%) had an absent/mild endoscopic activity (SES-CD < 7) and 10 (40%) a moderate/severe activity (SES-CD ≥ 7). The overall endoscopic characteristics are shown in Table 1.

### 3.3. Frequency of Dysplasia

Of the 105 pseudopolyp-like lesions analyzed, 18 (17%) were located in the right colon, 10 (9%) in the transverse colon, 52 (50%) in the left colon, and 25 (24%) in the rectum. Twenty-three of 105 pseudopolyp-like lesions (22%) had dysplastic foci. The histopathological analysis showed that 6 of these lesions (26%) were adenoma with low-grade dysplasia, 7 (30%) were defined as colitis-associated dysplasia (low-grade dysplasia), and 10 (44%) were hyperplastic with low-grade dysplasia. No pseudopolyp-like lesion was associated with high-grade dysplasia. Dysplastic lesions were solitary in 9 cases (39%) (2 adenomatous and 7 non-adenomatous), with inflamed mucosa around the polyp in 11 cases (3 adenomatous vs. 8 non-adenomatous), and/or an ulcerated surface in 9 (39%) cases (3 adenomatous and 6 non-adenomatous), and/or an undefined Kudos’ pit pattern in 3 (13%) cases (all of them adenomatous).

Three examples of pseudopolyp-like lesions (colitis-associated dysplasia, hyperplastic with dysplasia, and hyperplastic without dysplasia) are shown in Figure 1.

### 3.4. Clinical and Endoscopic Factors Associated with Dysplasia

The clinical and endoscopic characteristics of IBD patients with pseudopolyp-like lesions exhibiting either dysplasia or no dysplasia are shown in Table 1. Patients with dysplastic pseudopolyp-like lesions had a median age greater than that of patients with non-dysplastic pseudopolyp-like lesions. Moreover, patients with dysplastic pseudopolyp-like lesions had a more frequent history of colonic dysplasia and less exposure to immunosuppressors/biologics. Dysplasia was seen more frequently in pseudopolyp-like lesions located in the right colon and less frequently in those located in the left colon. Additionally, dysplastic pseudopolyp-like lesions had a median size greater than that of non-dysplastic pseudopolyp-like lesions and were more frequently solitary. No difference was seen between CD and UC patients, probably due to the small number of patients with CD. Clinical and endoscopic details of IBD patients with pseudopolyp-like lesions exhibiting either dysplasia or no dysplasia are shown in Table 2.

### 3.5. Predictive Factors of Dysplasia

The univariate analysis showed that the age of the patients (odds ratio (OR) 1.05, 95% CI 1.01 to 109, *p* = 0.007), previous history of dysplasia (OR 8.52, 95% CI 1.45 to 50.02, *p* = 0.014), the presence of a solitary lesion (OR 3.5, 95% CI 1.2 to 9.6), the size (OR 1.23, 95% CI 1.1 to 1.4, *p* = 0.0008), and right colonic location (OR 8.35, 95% CI 2.7 to 25.9, *p* = 0.0002) of the lesion were identified as risk factors of dysplasia (Table 3). In contrast, previous exposure to immunosuppressors and biologics (OR 0.28, 95% CI 0.11 to 0.75, *p* = 0.011) and left colonic location (OR 0.2, 95% CI 0.06 to 0.65, *p* = 0.003) were identified as protective factors. Multivariate analysis confirmed that the age of the patients (OR 1.1, 95% CI 1.02 to 1.12, *p* = 0.012), the size (OR 1.39, 95% CI 1.15 to 1.68, *p* = 0.0005), and right colonic location (OR 5.32, 95% CI 1.01 to 26.9, *p* = 0.04) of the lesions were independent predictors of dysplasia (Table 3). Previous exposure to immunosuppressors or biologics and left colonic location were inversely correlated to dysplasia (OR 0.11, 95% CI 0.02 to 0.52, *p* = 0.005, and OR 0.09, 95% CI 0.02 to 0.54, *p* = 0.0008, respectively) (Table 4). To establish which cut-off of the lesion size and age predicted the risk of dysplasia, two ROC curves were plotted. A lesion size > 5 mm and an age of 53 years had a sensitivity of 87% and 74% and a specificity of 63% and 57%, respectively, to identify dysplastic lesions (Figure 2) with a positive predictive value (PPV) of 39% and 32% and negative predictive value (NPV) of 94% and 94%, respectively.

## 4. Discussion

The term “pseudopolyps” has been applied to describe the presence of surviving islets of colonic mucosa between ulcers, which give the impression of a polyp during colonoscopy. Traditionally, pseudopolyps have been considered benign even though their presence has been independently associated with more severe disease activity, a greater need for treatment escalation, and escalation to biologic agents or surgery [23]. Consistently, in large tertiary care populations of IBD patients, a prior diagnosis of pseudopolyps was found to be an important predictive factor for CRC [11,13]. However, this association was not confirmed in other studies. For instance, in a retrospective analysis of data from two large independent surveillance cohorts, Mahmoud and collaborators showed that pseudopolyps were not associated with the development of CRC [24]. In line with this is also the study by Ryan Choi and collaborators who followed 987 patients with UC for a median of 13 years and showed that pseudopolyps were not associated with the development of CRC [25]. If the discrepancies among these studies reflect differences in the study’s design, the selection of the patients with or without exposure to drugs known to attenuate the risk of neoplastic transformation, and the duration of the follow-up, remains to be verified. It is also noteworthy that the accurate definition of pseudopolyps can be made difficult by concomitant polyps that appear endoscopically different from classical pseudopolyps, the presence of which could differently influence the risk of neoplastic transformation.

This study was undertaken to assess the frequency of dysplasia in pseudopolyp-like lesions evident during surveillance colonoscopy in patients with longstanding IBD. The 105 patients included had at least one pseudopolyp-like lesion, which exhibited endoscopic features difficult to be accurately interpreted and, therefore, was either biopsied or removed. We show that nearly one-fourth of such lesions had dysplastic foci at histology. One-fourth of the dysplastic lesions were adenomas, which appeared endoscopically with an undeterminable Kudo pit pattern. This finding is in line with data from previous studies indicating that the Kudo pit pattern classification alone is not sufficient to distinguish between adenomatous and hyperplastic polyp lesions in IBD [26]. We have previously shown that dysplastic lesions evidenced by DCE in IBD patients had more frequently a Kudo’s pit pattern III–IV, which is considered neoplastic in non-IBD patients, even though 18% of the dysplastic lesions had a Kudo pit pattern I–II [27]. 

In a recent review of the different histologic features associated with carcinoma in IBD, Gui and colleagues underlined that colitis-associated dysplasia exhibited a mixed/intermingled with inflammatory pseudopolyps and/or granulation tissue [28], revealing a histologic overlap and confirming the difficult to differentiate the lesions.

Notably, nearly half of the dysplastic lesions of the present study were seen in hyperplastic polyps. The malignant transformation of pseudopolyps is considered a rare event, and only a few cases of large pseudopolyps harboring carcinoma or dysplasia have been published [16,18]. However, analysis of 30 pseudopolyp samples resected from 30 IBD patients identified mutations in CDKN2A, tP53-exon7, and K-RAS genes in nearly 15% of the lesions [29]. Since such mutations are known to be protumorigenic, it is conceivable that some subsets of IBD-associated polypoid lesions can be dysplastic.

Multivariate analysis showed that the age of the patients and the size and right location of the pseudopolyp-like lesion were predictors of dysplasia, thus confirming and expanding on data from previous studies showing that the proximal location is an endoscopic characteristic predictive of dysplasia in longstanding IBD as evaluated by DCE in a real-life setting [30]. Our data also indicate that a pseudopolyp-like lesion with a size greater than 5 mm is suspicious of dysplasia and therefore should be treated as is. Previous exposure to immunosuppressors or biologics was inversely correlated to dysplasia, probably reflecting the positive impact of the drug-induced colitis attenuation on the risk of neoplastic transformation. 

Our study has some limitations, the most important of which is the fact that this was a retrospective study conducted on a relatively small number of pseudopolyp-like lesions. Nonetheless, the data used in this study relied on accurate documentation at the time of the surveillance procedures. The study included consecutively all patients with IBD referred for CRC screening in a tertiary center and was performed by expert endoscopists in surveillance colonoscopy.

Another limitation of this study was determined by possible selection bias due to the arbitrary endoscopist’s decision to take biopsies or remove only the greatest lesions. However, all the lesions included in our analysis were classified as pseudopolyp-like lesions and therefore the possibility of dysplasia was estimated very low during the procedure.

Although the surveillance colonoscopy was performed with different techniques (white light, NBI, DCE), the operators were aware of the nature of the study and performed the procedure more diligently than in routine clinical practice to limit the risk of evaluating differently the lesions.

We feel that our results deserve confirmation by prospective and standardized studies and need external validity before being generalized to the overall healthcare centers. It is also likely that the rate of dysplasia in our cohort is underestimated as most lesions were only biopsied and not removed, and the negativity of dysplasia in single biopsy samples does not exclude the possibility that the remaining portions of the pseudopolyps bear dysplastic foci.

To optimize the management of patients with pseudopolyps, future research on this clinical topic should aim to collect better documentation of the lesion’s characteristics and corresponding histology. Further work will also be needed to ascertain which molecular and genetic alterations promote dysplastic lesions, and whether this sequence of events could be stopped by drugs used in IBD (e.g., biologics and small molecules).

## 5. Conclusions

Our findings show that one-fourth of pseudopolyp-like lesions evident during surveillance colonoscopy in patients with longstanding IBD bear dysplastic foci and suggest treating such lesions properly.

## Figures and Tables

**Figure 1 cancers-15-03361-f001:**
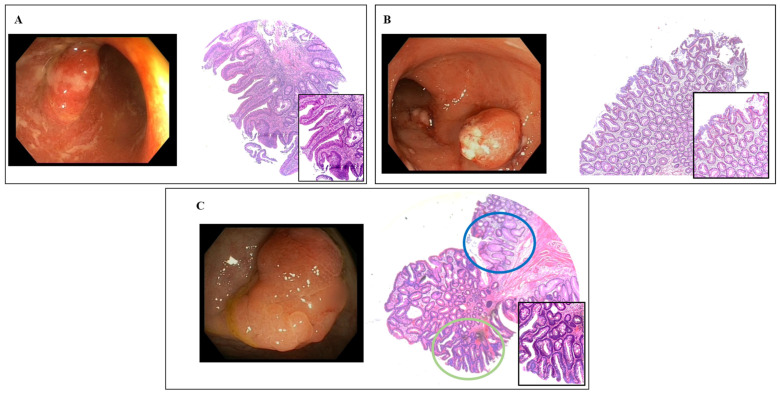
Representative endoscopic and histologic (hematoxylin & eosin staining) pictures of pseudopolyp-like lesions: Panel (**A**) colitis-associated dysplasia (×10; insert ×20); Panel (**B**) hyperplastic without dysplasia (×4; insert ×20); Panel (**C**) hyperplastic features without (blue circle) and with dysplastic foci (green circle) (×4; insert ×20).

**Figure 2 cancers-15-03361-f002:**
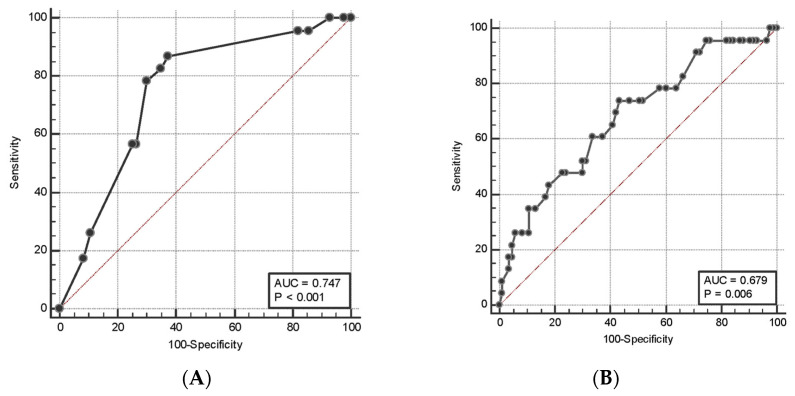
Receiver operating characteristic curves for dysplasia based on the lesion’s size (panel (**A**)) and age (panel (**B**)).

**Table 1 cancers-15-03361-t001:** Demographic, clinical and endoscopic characteristics of 105 included patients.

Characteristics of Patients	
**Age (years)**	
*median (range)*	54 (26–88)
**Sex, *n* (%)**	
Male	70 (67)
**Disease duration**	
*Median (range)*	192 (90–504)
**Age at disease onset**	
*Median (range)*	35 (12–67)
**UC, *n* (%)**	80 (76)
*E2*	52 (65)
*E3*	28 (35)
**CD, *n* (%)**	25 (24)
*L2*	19 (76)
*L3*	6 (24)
**IBD-U, *n* (%)**	0
**Family history of CRC, *n* (%)**	17 (16)
**Concomitant PSC, *n* (%)**	0
**Previous therapy with IMMs or biologics, *n* (%)**	65 (62)
**Ongoing therapy with ISS or biologics, *n* (%)**	78 (74)
**Smoking habit, *n* (%)**	
*Yes*	11 (10.5)
*No*	40 (3)
*Former*	54 (51.5)
**Previous dysplasia, *n* (%)**	
*Yes*	6 (6%)
**Clinical activity**	
Partial mayo score	
*Median (range)*	2 (0–9)
Harvey–Bradshaw Index	
*Median (range)*	3 (2–7)
**Endoscopic activity**	
Mayo UC score, (*n*)	
0, *n* (%)	33 (31)
1, *n* (%)	8 (8)
2, *n* (%)	17 (16)
3, *n* (%)	22 (21)
SES-CD, (*n*)	
*Median, range*	7 (1–28)
**BBPS**	
*Median, range*	8 (3–9)
**Type of endoscopy, *n* (%)**	
Chromoendoscopy (virtual or dye)	54 (51)
White light	51 (49)

IBD-U = IBD-unclassified; CRC = colorectal cancer; PSC = primary sclerosing cholangitis; ISS = immunosuppressors; BBPS = Boston Bowel Preparation Score.

**Table 2 cancers-15-03361-t002:** Clinical and endoscopic details of IBD patients with pseudopolyp-like lesions exhibiting either dysplasia or no dysplasia.

	Dysplasia *n* = 23	No Dysplasia *n* = 82	*p* Value
** *Clinical details* **			
Age-years			
Median (range)	60 (26–80)	50 (30–80)	**0.004**
Sex, *n* (%)			
*Male*	17 (74)	53 (65)	0.46
IBD type, *n* (%)			
*Ulcerative Colitis*	18 (78)	62 (76)	0.79
*Crohn’s Disease*	5 (22)	20 (24)	
Family history of colorectal cancer, *n* (%)	6 (26)	11(13)	0.35
Disease duration—months			
Median (range)	180 (95–384)	198 (90–504)	0.81
Current smoking, *n* (%)			
yes	3 (13)	8 (9)	0.79
Previous dysplasia, *n* (%)	4 (17)	2 (2)	**0.02**
Previous ISS/biologics	12 (52)	66 (80)	**0.015**
** *Endoscopic Details* **			
Inflammation score, median (range)			
*Mayo UC score*			
*- Remission/mild activity (0–1)*	10 (44)	31 (38)	0.79
*- Moderate/severe activity (2–3)*	8 (35)	31 (38)	
*SES-CD*			
*- Remission/mild activity (<7)*	4 (17)	10 (12)	0.34
*- Moderate/severe activity (≥7)*	1 (4)	10 (12)	
Procedures with adequate bowel preparation (BBPS ≥ 6), *n* (%)	16 (70)	52 (63)	0.63
Type of endoscopy			
Chromoendoscopy (virtual or dye)	10 (43)	44 (54)	0.48
White light	13 (57)	38 (46)	

ISS = immunosuppressors; BBPS = Boston Bowel Preparation Score.

**Table 3 cancers-15-03361-t003:** Details of the pseudopolyp-like lesions exhibiting either dysplasia or no dysplasia.

	Dysplasia *n* = 23	No Dysplasia *n* = 82	*p* Value
Location			
- Right colon vs. other segments	10 (43)	8 (10)	**0.0006**
- Transverse colon vs. other segments	2 (9)	8 (10)	0.1
- Left colon vs. other segments	8 (35)	44 (53)	**0.004**
- Rectum vs. other segments	3 (13)	22 (27)	0.26
Size mm, median (range)	10 (3–15)	5 (3–13)	**<0.0001**
Surface ulceration, *n* (%)	9 (39)	24 (29)	0.44
Number, *n* (%)			
Solitary	9 (39)	12 (15)	**0.04**
Multiple	14 (61)	70 (85)	
Inflamed surrounding mucosa, *n* (%)	11 (48)	36 (44)	0.81

**Table 4 cancers-15-03361-t004:** Predictive clinical and endoscopic features associated with risk of dysplasia in pseudopolyp-like lesions.

Risk Factors	Univariate Analysis		Multivariate Analysis	
	OR (95% CI)	*p* Value	OR (95% CI)	*p* Value
Age, years	1.05 (1.01 to 1.09)	**0.007**	1.1 (1.02 to 1.12)	**0.012**
IBD type: ulcerative colitis vs. Crohn disease	0.82 (0.27 to 2.48)	0.72		
Disease duration	0.98 (0.99 to 1.01)	0.65		
Previous dysplasia	8.52 (1.45 to 50.02)	**0.014**	9.01 (0.77 to 10.5.2)	0.07
Previous ISS/biologics	0.28 (0.11 to 0.75)	**0.011**	0.11 (0.02 to 0.52)	**0.005**
Moderate/severe endoscopic activity	0.55 (0.21 to 1.42)	0.22		
Use of chromoendoscopy	0.67 (0.26 to 1.72)	0.41		
Lesion’s size	1.23 (1.1 to 1.4)	**0.0008**	1.39 (1.15 to 1.68)	**0.0005**
Lesion location				
- Cecum/right colon	8.35 (2.7 to 25.9)	**0.0002**	5.32 (1.01 to 26.9)	**0.04**
- Transverse colon	1.03 (0.19 to 5.3)	0.96		
- Left colon/sigmoid colon	0.2 (0.06 to 0.65)	**0.003**	0.09 (0.02 to 0.54)	**0.0008**
- Rectum	1.38 (0.49 to 3.82)	0.54		
Presence of inflamed mucosa around the lesion	1.19 (0.47 to 3.02)	0.7		
Ulcerated surface	1.58 (0.6 to 4.14)	0.35		

ISS = immunosuppressors.

## Data Availability

The data that support the findings of this study are available from the corresponding author (G.M.), upon reasonable request.

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
