# Peer review of "Frequency and Predictors of Dysplasia in Pseudopolyp-like Colorectal Lesions in Patients with Long-Standing Inflammatory Bowel Disease"

_cancers, 2023, doi:10.3390/cancers15133361_

Round 1

Reviewer 1 Report

I appreciate the authors’ complex work (and, mainly their idea) which led to important findings regarding dysplasia in colorectal pseudopolyp-like lesions (found in area with colitis), in patients with more than 8 years of IBD course (frequency and predictor factors). Almost a quarter of the polyps with dysplasia (even if low grade – no patients with high grade) represents something meaningful and we should be aware of this in our practice. Inasmuch as I appreciate their manuscript, I have some comments/questions:

*Main concerns:

1. The polyps were either biopsied or resected. I saw that this was considered a limitation of the study by the authors as well, but it is a major aspect, most important. Lines 255-256 – the authors wrote: “It is also likely that the rate of dysplasia in our cohort is underestimated as most lesions were only biopsied and not removed”. Please insert how many polyps were just biopsied. I did not find it anywhere.

2. The Authors wrote: “In case of multiple lesions, the pseudopolyp with the greater size was biopsied or removed.” Only one lesion from each patient, which was considered to have the largest dimensions. What about smaller lesions? If a patient had a large polyp and only that one was removed/biopsied, how can we be sure that dysplasia was not present in a smaller one? (still > 5 mm, even if according to Table 2 – a polyp of just 3 mm had dysplasia). Therefore, many other polyps with dysplasia could have been omitted. This aspect should be approached in Discussion, as it seems that size just > 5 mm was predictive for dysplasia. And this is not a large polyp. Another possibility for bias.

3. The endoscopy technique was not the same, as mentioned in lines 88-89: “technique of mucosa evaluation [i.e. high definition white light, narrow binding imaging (NBI) or dye-chromoendoscopy (DCE) with methylene blue or indigo carmine]. This could introduce another bias.

 * Other comments:

1. Title: I would suggest to include “colorectal”, as the manuscript refers only to pseudopolyp-like lesions in patients with colitis and not any other location (I mean for patients with Crohn’s disease). (“Frequency and predictors of dysplasia in pseudopolyp-like colorectal lesions in patients with long-standing inflammatory bowel disease). Same in “Simple Summary”.

2. Abstract:

a. Please mention clearly the “aim” of the study, not what you “evaluated”.

b. Please include the study period, whether it was a single/multicentre study (from the main text it appears single-centre), the fact that it was retrospective and the mean duration of IBD (since it was considered longstanding) – mentioned in the main text.

c. Also, it would be interesting to mention here also the number of UC and CD patients (from the main text, it appears that no IBDU patient was included) and whether there were any differences in outcomes. From the main text, it appears that there were no differences between CD and UC dysplastic lesions (but also the number of CD patients is much smaller). Please include, as this finding is important.

d. So, there were 105 patients out of 200 enrolled? Why were 200 enrolled? While reading further, from the main text, it appears that only 105 patients had “pseudopolyps”. Then, please include (enroll) only 105 patients, as per the inclusion criteria written in the main text (criterium nr. 4). Writing 200 ENROLLED patients makes no sense, since only 105 patients met the inclusion criteria (having pseudo-polyps). Please revise.  

e. “Multivariate analysis showed that the age of the patients” – please be more specific and insert also the result from the main text.

3. Introduction: generally good, but based on old references. Please insert more recent ones.

a. Line 46: “Age at diagnosis” – please be more specific, as data are available.

b. It would be great to include the Kudo pit pattern classification, for those who are not aware of. Or, at least, please insert the original reference.

c. Lines 64-65: “This study aimed to assess the frequency of and predictors of dysplasia in IBD-associated pseudopolyp-like lesions” – please insert, as mentioned above “colorectal”, before lesions, as your study involved only patients with colitis and not any other location, like in CD. Also, please mention “longstanding IBD”.

4. Materials and Methods: Study population and data collection:

a. The study period is missing.

b. The authors mentioned as inclusion criteria “confirmed IBD of any duration, in the patients with a concomitant PSC”. I did not find in Results any patient with PSC. Were there any? Just wondering. If not, it should be mentioned.

c. Line 90: please correct “f”.

d. Lines 85-86: The authors wrote: “disease activity (defined using the Mayo UC score and SES-CD), but in the lines 126-128, it appears that: “Among CD patients, 20 (80%) were either in clinical remission or mild activity (Harvey Bradshaw Index < 8), and 5 (20%) had moderate/severe activity (Harvey Bradshaw Index ≥ 8). Please mention Harvey Bradshaw Index in Methods as well (for clinical activity).

e. Line 86: “quality of bowel preparation” – please insert reference, as in Results you introduced BBPS (Boston bowel preparation scale) – not defined before.

5. Results:

a. “Patients’ characteristics” – could be nicely organized in a table, together with “Endoscopic characteristics”. I appreciate the data in the presented tables, but I consider this one would be helpful, too.

b. Figure 1: Please enlarge microscopical images, write magnification and mention that it was hematoxylin-eosin stain.

6. Discussion:

a. Please insert the title of this paragraph, before line 200.

b. Line 221: Please correct: not 200 patients enrolled, but 105.

c. Please insert future directions for research. Please consider also inclusion of other techniques (antigens, genetics etc).

7. References are old. As mentioned, there are no data about the period the study was performed, but there are many new references available. Except for the authors’ own study, dated 2022, many other references are quite old. Sure, they are good and correctly inserted, but the recent ones have to be mentioned and commented on, too.

Thank you.

Minor editing of English language is required.

Reviewer 2 Report

Working within the limitations of being a retrospective study and comparatively limited numbers, the authors have again politely put into question the issues of the potential of inflammatory polyps developing into colonic cancer.

Combining inflammatory induced dysplasia within polyps whose inflammatory genesis is the product of divergent etiologies presents a secondary issue not fully addresses. The pathogenesis of CD is not the pathogenesis of UC. Ileocecal and right colonic lesions are more likely a response to inflammation secondary to MAP/cytokine interaction. What elicits an inflammatory response in what is probably a collection of divergent processes lumped under the label of Unci’s an open issue.

Having the greatest probability of being the consequence of a single mechanism, a tighter focus needs to be focused on left-sided CRC. Given that dysplastic changes within inflammatory polyps here, the question is why. Are the changes secondary to accelerated mucosal regeneration or a response to a concentrated exposure to dietary introduced carcinogens. “Nearly half of the dysplastic lesions were seen in hyperplastic polyps”.

The discussion induced mutagenesis is worthy of expansion.

More discussion on WHY would enhance the manuscript.

Overall, the authors did a creditable job dealing with the imposed limitations.

Reviewer 3 Report

Very important subject

the study was retrospective but with a good design

the results were clear and interesting

there is not a very substantial message for the readers 

we are looking forward for prospective study with better applicability in the current practice

Reviewer 4 Report

I would recommend that the methodology be improved 

All procedures will be the same protocol ( NBI)

Better define what is pseudopolyp  vs.  mucosal irregularity/DALM

Picture every lesion, for acceptance by endoscopic review 

All  lesions be removed

Dedicated two pathologists review 

The inttesity of inflmmation be validated 

Round 2

Reviewer 1 Report

The manuscript was revised and corrected. However, there are some improvements to be made.

A. ABSTRACT

Almost none of my suggestions for an improved Abstract was taken into consideration. The Abstract represents the essence of the whole paper. If paramount important data are not included, then readers may not be interested in reading the paper, thus not improving their practice and this will be against the well-being of our patients. This is why I recommended the changes. Examples below:

1.I wrote: Please mention clearly the “aim” of the study, not what you “evaluated”.

Response: We made it clearer in the abstract section (lines: 67-71).

***However, no changes were made in the Abstract. Moreover, lines 67-71 do not belong to the Abstract section.

2. I wrote: Please include the study period, whether it was a single/multicentre study (from the main text it appears single-centre), the fact that it was retrospective and the mean duration of IBD (since it was considered longstanding) – mentioned in the main text.

Response: This information was added in the material and methods section (paragraph 2; line 74 and paragraph 3 lines 128-129).

***However, the Abstract was not corrected.

3. I wrote: Also, it would be interesting to mention here also the number of UC and CD patients (from the main text, it appears that no IBDU patient was included) and whether there were any differences in outcomes. From the main text, it appears that there were no differences between CD and UC dysplastic lesions (but also the number of CD patients is much smaller). Please include, as this finding is important.

Response:  Lines 125-127: as reported in the manuscript, 80 patients had UC and 25 CD. No patient had IBD-U. We added this information in the text (line 128). No difference was found between UC and CD, probably due to the small number of patients with CD (lines 189-190).

***However, nothing was inserted in the Abstract.

4. I wrote: So, there were 105 patients out of 200 enrolled? Why were 200 enrolled? While reading further, from the main text, it appears that only 105 patients had “pseudopolyps”. Then, please include (enroll) only 105 patients, as per the inclusion criteria written in the main text (criterium nr. 4). Writing 200 ENROLLED patients makes no sense, since only 105 patients met the inclusion criteria (having pseudo-polyps). Please revise

Response: We made the suggested changes (results section).

***However, nothing was changed in the Abstract. This is very important, as it conflicts with the inclusion /enrolment criteria.

5. I wrote: “Multivariate analysis showed that the age of the patients” – please be more specific and insert also the result from the main text.

Response: This data was inserted in the results section (lines 200 – 201)

***Nothing was changed in the Abstract.

All these changes should be made to the ABSTRACT section (paragraph).

B. Line 89 – “PSC” was used for abbreviation, while in Table 1 it was written “CSP”. Please correct, otherwise it appears confusing for readers.

C. Table 1 – please define abbreviations, including “CSP” or(?) “PSC”, “C-UNC” and “ISS”. I guess C-UNC - means IBD unclassified (that would be IBD-U, according to the abbreviation used nowadays). In fact, also the Authors used “IBD-U”, in line 127. “ISS” – immunosuppressors probably, but in Tables 3 and 4 – the abbreviation “IMM” was used (probably from immunomodulators). Please decide and unify, otherwise it appears confusing.

Thank you

Generally good, only minor revision is required.

Reviewer 4 Report

As I wrote before,  there is concern about the study design
